Ecological and Evolutionary Science
# *Alteromonas* Myovirus V22 Represents a New Genus of Marine Bacteriophages Requiring a Tail Fiber Chaperone for Host Recognition

Rafael Gonzalez-Serrano,[a] Matthew Dunne,[b] Riccardo Rosselli,[c,e] Ana-Belen Martin-Cuadrado,[d] Virginie Grosboillot,[b] Léa V. Zinsli,[b] Juan J. Roda-Garcia,[a] Martin J. Loessner,[b] Francisco Rodriguez-Valera[a,e]

[a]Evolutionary Genomics Group, Universidad Miguel Hernández, San Juan de Alicante, Spain
[b]Institute of Food, Nutrition and Health, ETH Zurich, Zurich, Switzerland
[c]NIOZ Royal Netherlands Institute for Sea Research, Department of Marine Microbiology and Biogeochemistry, Utrecht University, Den Burg, The Netherlands
[d]Universidad de Alicante, San Vicente del Raspeig, Spain
[e]Laboratory for Theoretical and Computer Studies of Biological Macromolecules and Genomes, Moscow Institute of Physics and Technology, Dolgoprudny, Russia

Rafael Gonzalez-Serrano and Matthew Dunne contributed equally to this work. Author order was determined by mutual agreement.

**ABSTRACT** Marine phages play a variety of critical roles in regulating the microbial composition of our oceans. Despite constituting the majority of genetic diversity within these environments, there are relatively few isolates with complete genome sequences or in-depth analyses of their host interaction mechanisms, such as characterization of their receptor binding proteins (RBPs). Here, we present the 92,760-bp genome of the *Alteromonas*-targeting phage V22. Genomic and morphological analyses identify V22 as a myovirus; however, due to a lack of sequence similarity to any other known myoviruses, we propose that V22 be classified as the type phage of a new *Myoalterovirus* genus within the *Myoviridae* family. V22 shows gene homology and synteny with two different subfamilies of phages infecting enterobacteria, specifically within the structural region of its genome. To improve our understanding of the V22 adsorption process, we identified putative RBPs (gp23, gp24, and gp26) and tested their ability to decorate the V22 propagation strain, *Alteromonas mediterranea* PT11, as recombinant green fluorescent protein (GFP)-tagged constructs. Only GFP-gp26 was capable of bacterial recognition and identified as the V22 RBP. Interestingly, production of functional GFP-gp26 required coexpression with the downstream protein gp27. GFP-gp26 could be expressed alone but was incapable of host recognition. By combining size-exclusion chromatography with fluorescence microscopy, we reveal how gp27 is not a component of the final RBP complex but instead is identified as a new type of phage-encoded intermolecular chaperone that is essential for maturation of the gp26 RBP.

**IMPORTANCE** Host recognition by phage-encoded receptor binding proteins (RBPs) constitutes the first step in all phage infections and the most critical determinant of host specificity. By characterizing new types of RBPs and identifying their essential chaperones, we hope to expand the repertoire of known phage-host recognition machineries. Due to their genetic plasticity, studying RBPs and their associated chaperones can shed new light onto viral evolution affecting phage-host interactions, which is essential for fields such as phage therapy or biotechnology. In addition, since marine phages constitute one of the most important reservoirs of noncharacterized genetic diversity on the planet, their genomic and functional characterization may be of paramount importance for the discovery of novel genes with potential applications.

**KEYWORDS** *Alteromonas*, phage V22, receptor binding protein, tail fiber, tail fiber chaperone, *Myoalterovirus*, bacteriophage, host recognition

Address correspondence to Matthew Dunne, mdunne@ethz.ch, or Francisco Rodriguez-Valera, frvalera@umh.es.

Prokaryotic viruses (phages) are the most abundant and genetically diverse entities in the oceans (1, 2). As the predominant predators of bacteria, phages play critical roles throughout the marine ecosystem, for instance, regulating bacterial diversity through host cell lysis (3) or horizontal gene transfer of heritable information such as virulence factors and antimicrobial resistance genes (2, 4). Marine phages are also major contributors to global biogeochemical cycling of carbon, nitrogen, and phosphorus (5) and are involved in the recycling of organic matter through prokaryotic cell lysis in a process known as the viral shunt (6, 7). Despite their ecological relevance, there are relatively few marine phages with complete genome sequences available to expand our understanding of marine evolution (8). Furthermore, as they constitute the biggest reservoir of noncharacterized genetic diversity on Earth (2), genomic and functional characterization of marine phages could aid the discovery of new genes with the potential for application in biological research.

Members of the genus *Alteromonas* populate the oceanic euphotic and aphotic zones and have been isolated all around the world (9, 10). This heterotrophic bacterium plays an important role in marine organic carbon and nitrogen recycling (11), and its genomes have been analyzed for years using comparative genomics in order to better understand its genomic composition to generate evolutionary models (12). It has been shown that some isolates of *Alteromonas* can synthesize exopolysaccharides (EPS) useful for production of colloidal suspension of silver nanoparticles (AgNPs) (13, 14), which are used in nanomedicine, pharmaceutical sciences, and biomedical engineering (15, 16). The first *Alteromonas* phages characterized were isolated from western Mediterranean coastal waters and identified as a new genus within the *Podoviridae* family (17). An *Alteromonas*-targeting *Podoviridae* phage isolated from the North Sea (18) has been used as a biological tracer in hydrological transport studies (19), and finally two *Alteromonas* phages belonging to the *Siphoviridae* family have been isolated from the western Yellow Sea (20, 21). These (seven in total) are the only *Alteromonas* phages isolated and characterized to date.

Metagenomics has rapidly expanded our understanding of the viral "dark matter" and uncovered numerous new single-stranded RNA and DNA phage genomes and their infection mechanisms (22, 23). Nevertheless, the majority of characterized marine phages are tailed phages with double-stranded DNA (dsDNA) genomes belonging to the order *Caudovirales* (24). Despite the variable morphology among the *Caudovirales* families, these phages typically mediate host adsorption via receptor binding proteins (RBPs) located on a baseplate structure at the distal end of their tails (24). RBPs are identified as globular tailspike proteins or tail fibers that can recognize a wide spectrum of host exposed receptors such as bacterial appendages (pilus or flagellum), lipopolysaccharides (LPS), or outer membrane proteins (25–27). RBPs present high genetic plasticity, which is an essential feature for phage adaptation to novel or evolved hosts (24, 28). Indeed, even when phage tails have similar morphologies, it is unusual that they recognize similar targets, explaining the high specificity observed in phage-host recognition systems (27). In this sense, RBPs can be exploited, for example, as tools for bacterial diagnostics (29–33).

The correct folding and maturation of these complex trimeric structures often require phage-encoded fiber chaperones (34–37). For instance, trimerization and correct folding of the receptor-binding distal tip (gp37) of the phage T4 long tail fibers (LTF) use two chaperones, gp57A and gp38, that are not incorporated into the final fiber complex (38). The gp57A chaperone is also required for correct folding of the LTF proximal segment (gp34) (39) and the T4 short tail fiber (gp12; STF) (34, 40). Similar fiber accessory proteins remain bound to the tail fiber after production to assist with host recognition, such as the tail fiber accessory protein (Tfa) of the phage Mu tail fiber or the gp38 adhesin that attaches to the distal tip of the *Salmonella* phage S16 tail fiber (37, 41). Due to their genetic plasticity and role in determining host range, RBPs and their corresponding chaperones are ideal model systems for studying viral evolution (42, 43).

Here, we report the isolation and genomic characterization of a novel *Alteromonas*

**TABLE 1** Host range analysis of phage V22

| *Alteromonas* sp. and strain | Infection[a] | GenBank accession no. |
|---|---|---|
| *A. mediterranea* | | |
| PT11 | + | NZ_CP041169 |
| PT15 | − | NZ_CP041170 |
| CH17 | − | NZ_CP046670 |
| DE1 | − | NC_019393 |
| U4 | − | CP004849 |
| U7 | − | NC_021717 |
| U8 | − | CP004852 |
| UM7 | − | NC_021713 |
| UM8 | − | CP013928 |
| UM4b | − | NC_021714 |
| | | |
| *A. macleodii* | | |
| AD45 | − | NC_018679 |
| Te101 | − | NZ_CP018321 |
| EZ55 | − | SAMEA5587614 |
| MIT1002 | − | NZ_JXRW00000000 |
| HOT1A3 | − | NZ_CP012202 |

[a]+, susceptible; −, not susceptible.

myovirus, vB_AmeM_PT11-V22 (V22) isolated from the Mediterranean Sea. As V22 has no sequence similarity to any other known *Myoviridae* besides an uncultured marine phage assembled from a viral metagenome, named GOV_bin_2917 (44), we propose the creation of the new genus *Myoalterovirus* within the *Myoviridae* family. In addition, we identify the V22 RBP (gp26) and show how host recognition by this new type of phage-encoded RBP is possible only after coexpression with an intermolecular chaperone (gp27).

## RESULTS

**V22 is a narrow-host-range myovirus.** Phage V22 was identified as a narrow-host-range phage due to its ability to form plaques (clear with turbid halos) only on the propagation strain PT11 when tested against a broad library of 15 different *Alteromonas* strains, including 10 from the species *Alteromonas mediterranea* (Table 1). V22 was further identified as a member of the *Myoviridae* family due to the presence of noncontracted and contracted tails using transmission electron microscopy (TEM) (Fig. 1A and B).

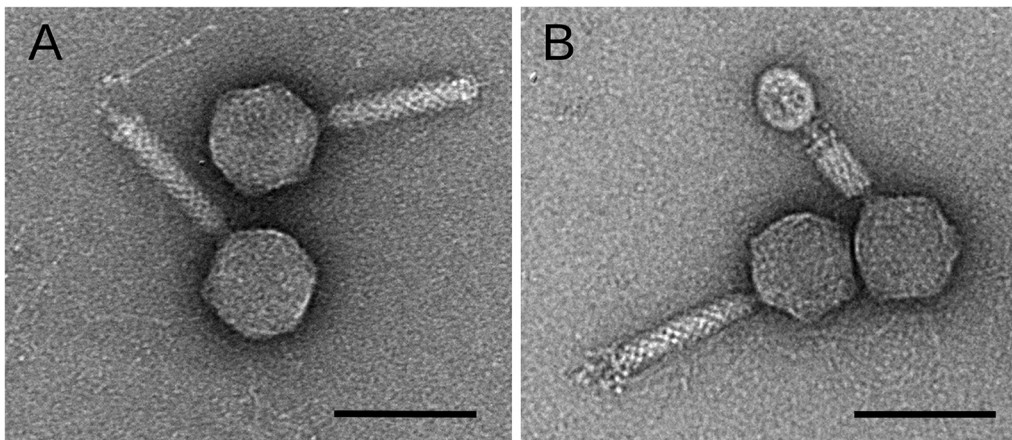

**FIG 1** Transmission electron microscopy of phage V22. The morphology of V22 is distinctly myoviral featuring noncontracted (A) and contracted (B) tails. V22 has an icosahedral capsid (ø79 ± 5 nm) and a noncontracted tail length of 121 ± 11 nm with a fiberless baseplate complex (20 ± 3.7 nm height). Dimensions calculated as mean ± SD, n = 6. Bar, 100 nm.

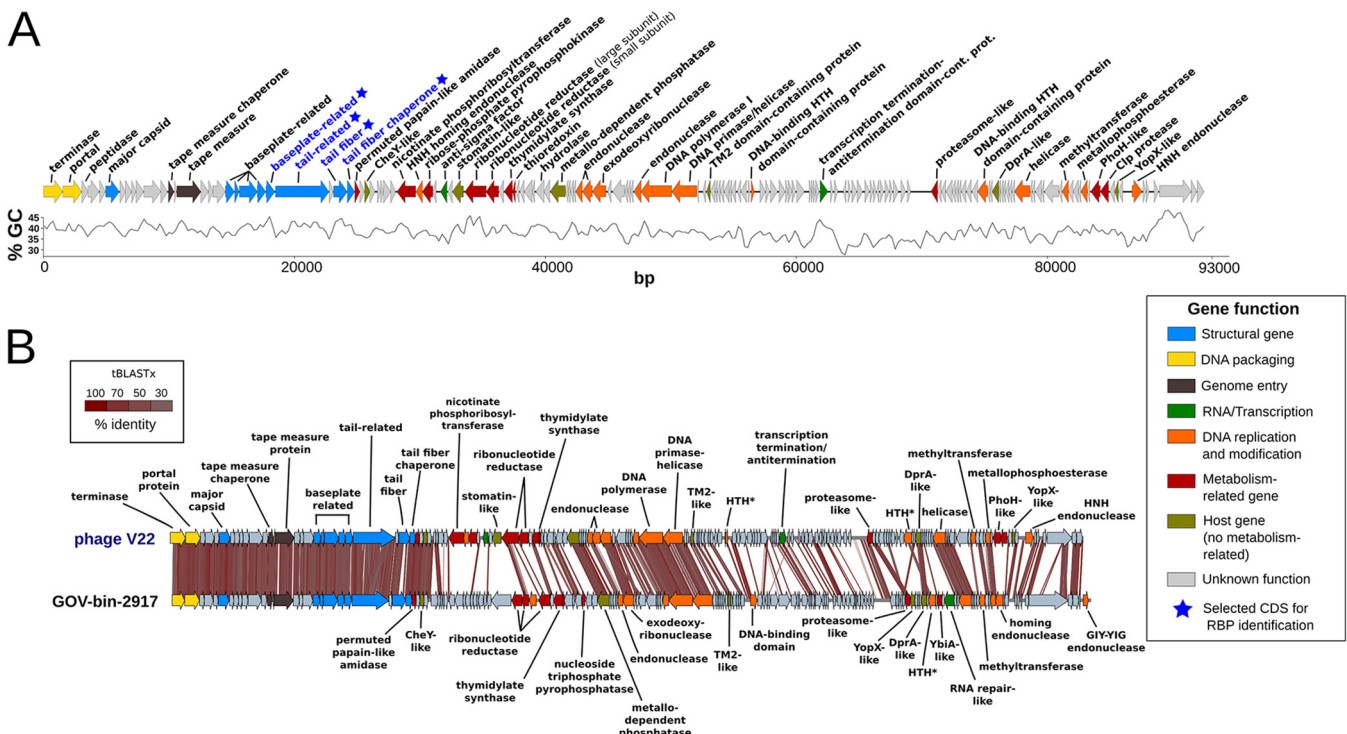

**FIG 2** V22 genome and alignment with the MAG GOV_bin_2917. (A) The genome of V22 is annotated and colored based on known molecular functions of identified genes with percent GC content for the entire genome displayed underneath. Blue stars indicate genes selected for investigation as putative RBPs as described below. Metabolism-related genes (in dark red) include class I AMGs and viral metabolism-related genes. Host genes (in green) include all class II AMGs. (B) Whole-genome comparison of V22 (92,760 bp; 157 predicted genes) and the MAG GOV_bin_2917 (93,482 bp; 161 predicted genes) using tBLASTx with 30% minimal identity on 10-bp minimum alignments. Genomes are aligned starting from their terminase genes. Homology is shown with red-shaded zones (scale indicates percent identity). HTH, DNA-binding helix-turn-helix domain-containing protein.

Sequencing of V22 revealed a double-stranded DNA 92,760-bp genome and a G+C content of 38.4% evenly distributed across the genome except for a single region between bp 52,208 and 70,988 bp where G+C content dropped to 35.6% (Fig. 2A). By comparison, the propagation strain PT11 (isolated in the same sampling campaign as V22) has a G+C content of 44.8% (see Table S1 in the supplemental material). The V22 genome has a coding density of 90%, consisting of 157 predicted coding DNA sequences (CDSs; mostly referred to as "gene products" [gp] in this study) with a near-even split of leading-strand and lagging-strand transcriptions (46% and 54%, respectively). No tRNA genes or secondary structures including noncoding RNAs (ncRNAs) were identified, and no integrase or other lysogeny-related genes could be found in the V22 genome, suggesting a strictly lytic life cycle.

Putative functions could be assigned to 46 CDSs (29.3%) due to significant sequence similarity to protein sequences and/or known protein domains (Table S2). For instance, two genes involved in DNA packaging could be identified: gp1, annotated as the large subunit terminase (TerL), showed 62% similarity to the terminase of the *Rheinheimera* phage Barba19A, and gp2, annotated as the portal protein, showed 60% similarity to the portal protein of *Vibrio* phage qdvp001. Unlike in most phages, which contain large and small terminase subunits (TerL and TerS) (45), we could not identify a putative TerS within the V22 genome. The identification of only a single TerL protein has been reported for other phages, including Felix O1, *Listeria* phage P100, or phage ϕ29 (45–47), and suggests that an alternative DNA packaging mechanism could exist for these phages that would require further investigation. Seven auxiliary metabolic genes (AMGs) were identified in the V22 genome, one from class I AMGs (a PhoH-like protein) and six from class II AMGs (e.g., a CheY, a stomatin, or a YopX-like proteins) (Table 2); all are potentially involved in the enhancement of host functionality to improve viral propagation (6, 48). Furthermore, nine genes associated with nucleotide metabolism or

**TABLE 2** Putative AMGs and viral metabolism-related genes identified in phage V22

| Coding DNA sequence | Annotation | Type of gene | Function |
|---|---|---|---|
| CDS-28 | Permuted papain-like amidase | Viral metabolism-related | Protein catabolism |
| CDS-30 | Histidine kinase/response regulator (CheY-like) | Class II AMG | Signaling |
| CDS-36 | Nicotinate phosphoribosyltransferase | Viral metabolism-related | Nucleotide metabolism |
| CDS-38 | Ribose phosphate pyrophosphokinase | Viral metabolism-related | Nucleotide metabolism |
| CDS-42 | Stomatin-like | Class II AMG | Quality control of membrane proteins |
| CDS-43 | Ribonucleotide reductase (large subunit) | Viral metabolism-related | Nucleotide metabolism |
| CDS-44 | Ribonucleotide reductase (small subunit) | Viral metabolism-related | Nucleotide metabolism |
| CDS-46 | Thymidylate synthase | Viral metabolism-related | Nucleotide metabolism |
| CDS-47 | Thioredoxin | Viral metabolism-related | Redox signaling for nucleotide metabolism |
| CDS-54 | Metallo-dependent phosphatase | Class II AMG | Cellular regulation and signaling |
| CDS-72 | TM2 domain-containing protein | Class II AMG | Transmembrane domain of unknown function |
| CDS-113 | Proteasome-like | Viral metabolism-related | Protein catabolism |
| CDS-126 | DprA-like | Class II AMG | Protection from incoming DNA (unclear) |
| CDS-146 | PhoH-like | Class I AMG | Phosphate recovery and metabolism |
| CDS-147 | Clp protease | Viral metabolism-related | Protein catabolism |
| CDS-149 | YopX-like | Class II AMG | Pathogenicity |

protein catabolism were identified (Table 2); however, according to the Brum and Sullivan classification (6), these genes are not considered AMGs as they are not involved in the improvement of host function (Fig. S1). Interestingly, none of the AMGs found in the V22 genome had been detected in any of the seven known *Alteromonas* phages.

**V22 is the type phage of the proposed genus *Myoalterovirus*.** A BLASTn comparison of the V22 genome against the nonredundant database at NCBI identified significant identity (76.49% identity, 18.3% query coverage; average values with alignments >1,000 bp) with only one other phage, a metagenome-assembled genome (MAG) of an undescribed and uncharacterized dsDNA virus named GOV_bin_2917 (GenBank accession no. MK892806), which was assembled from a viral metagenome sampled from the Indian Ocean during the *Tara* Oceans Expedition (44, 49). Genomic alignment between V22 and GOV_bin_2917 after tBLASTx analysis revealed the highest overall synteny and homology for proteins within the phage packaging and structural modules (gp1 to gp27) (Fig. 2B); for example, the portal protein (gp2) and major capsid protein (gp6) share 93.64% and 92.46% sequence similarity, respectively. The most variable protein within this region was gp26, which we later identified as encoding the V22 receptor binding protein (RBP), which still shares relatively high sequence similarity (59%) with its GOV_bin_2917 homolog. As noted by Adriaenssens and Brister (50), the Bacterial and Archaeal Viruses Subcommittee (BAVS) of the International Committee on the Taxonomy of Viruses (ICTV) considers phages sharing ≥50% nucleotide sequence identity as members of the same genus. The closest V22 relative identified was *Vibrio* phage 1.063.O._10N.261.45.C7 (GenBank accession number MG592441). sharing 32.9% average nucleotide identity (ANI) (see Fig. S2). Due to the lack of known myoviruses sharing an identity greater than 50% with V22, and the high ANI value (71.1%) found between V22 and GOV_bin_2917, we have proposed that both phages be considered members of a new genus called *Myoalterovirus* within the family *Myoviridae* (under ICTV review), featuring phage V22 as the type species.

**Phylogenetic analysis of V22.** To further understand the diversity of V22 and the proposed *Myoalterovirus* genus, a phylogenetic tree was generated using its terminase (gp1) and the 42 most similar terminase genes from other *Myoviridae* cultures and MAGs (Fig. 3). Three well-supported clades were identified: the *Ounavirinae* clade composed of 18 myoviruses infecting enterobacteria, the phylogenetically diverse *Vequintavirinae* clade featuring nine enterophages contained within the same monophyletic group (with 99% bootstrap value) and other marine phages, and the *Myoalterovirus*-containing clade. Enterobacterium phage T4 and seven other T4-like phages appeared as an outgroup. Phage V22 and its closest relative GOV_bin_2917 were grouped together into a diverse clade containing eight other marine phages with a 62% bootstrap value. The closest relatives to both *Myoalterovirus* representatives were

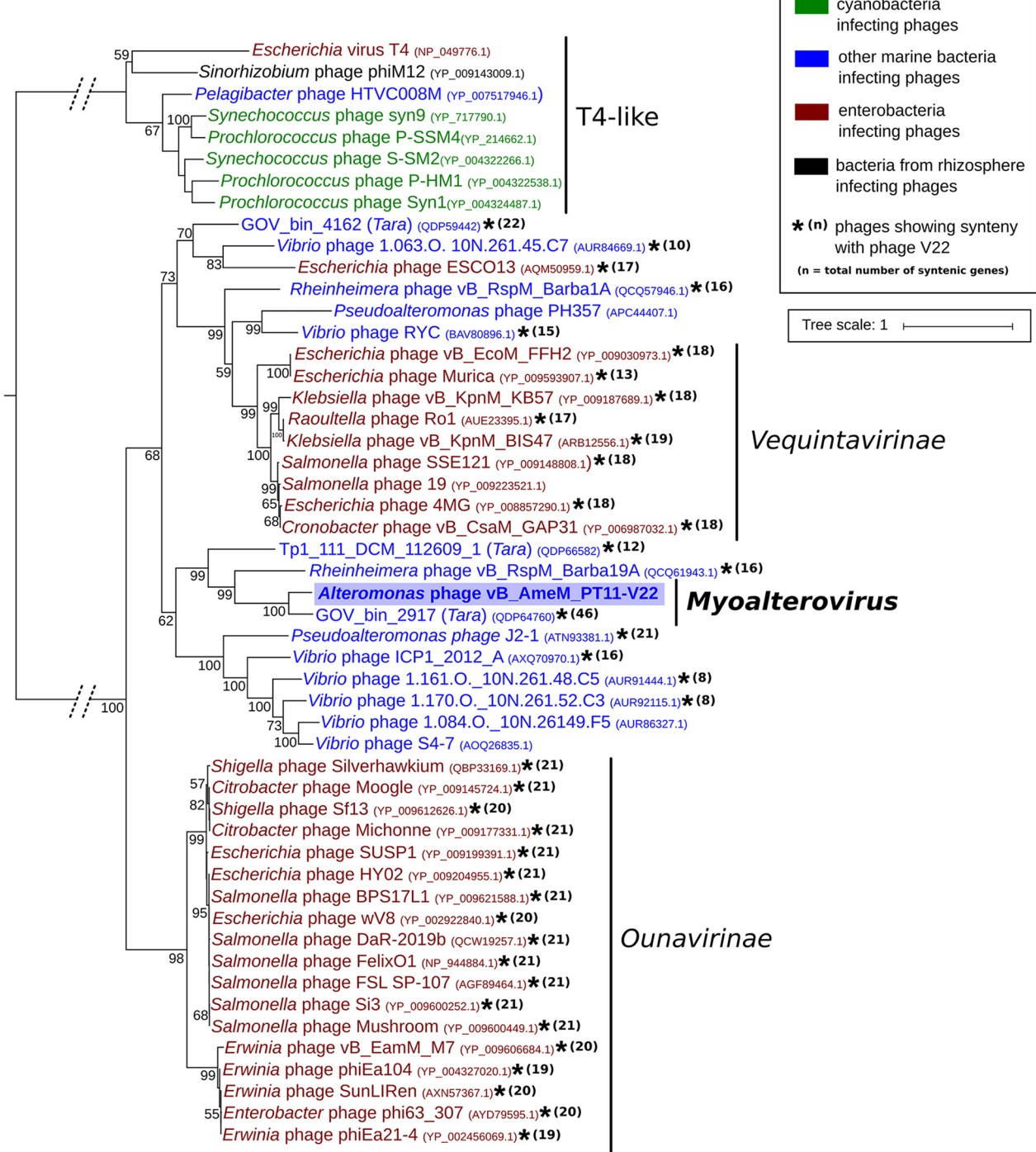

**FIG 3** Phylogenetic analysis of the V22 terminase. Fifty-one terminases were aligned using MUSCLE, and a tree was built using the maximum likelihood algorithm and 1,000 bootstrap replications with bootstrap values greater than 50 shown. Different colors indicate phages infecting different groups of bacteria (inset box). Dotted lines indicate a bigger phylogenetic distance between the T4-like monophyletic group and the rest of the clades (total distance estimate = 4.74). Asterisks indicate phages syntenic to V22. Synteny was considered when genomes presented at least five or more syntenic genes in a row located within the same genomic area and separated by a maximum of four nonsyntenic genes. *Vequintavirinae* and *Ounavirinae* are *Myoviridae* subfamilies. *n* = total number of syntenic genes across the whole genome.

the *Rheinheimera*-infecting phage Barba19A (51) and the MAG Tp1_111_DCM_112609 (44), with significant distance estimates of ca. 1.34 and 1.55, respectively. All together, these four phages clustered within the same monophyletic group with a bootstrap value of 99%. Although the *Vequintavirinae* and *Ounavirinae* monophyletic groups

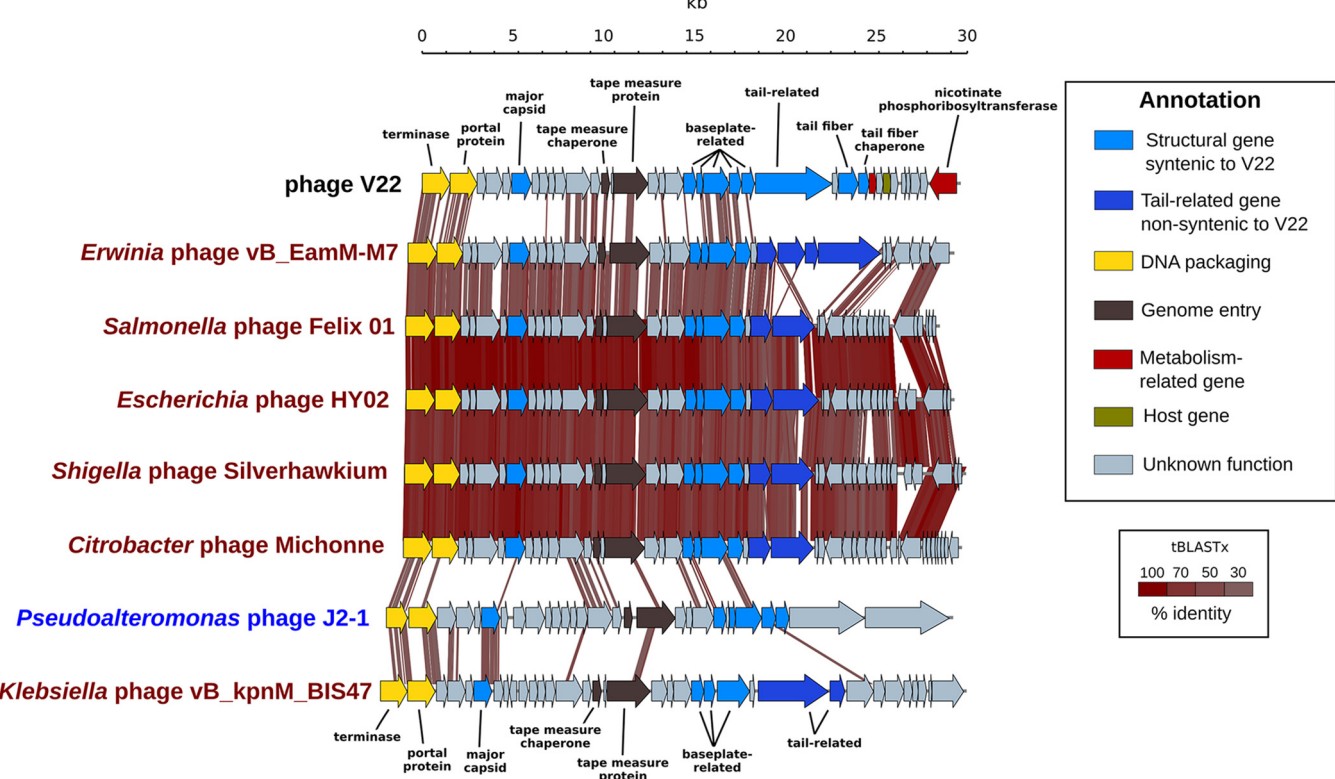

**FIG 4** Synteny within the structural modules of V22 and other myoviruses. A 30,000-bp region from the terminase (gp1) to a predicted nicotinate phosphoribosyltransferase (gp36) containing the V22 structural genes is shown aligned to six phages infecting enterobacteria (in red) (GenBank accession numbers: M7, NC_041978; Felix O1, NC_005282; HY02, NC_028872; Silverhawkium, MK562505; Michonne, NC_028247; BIS47, KY652726) and the marine *Pseudoalteromonas* phage J2-1 (in blue) (GenBank accession no. MF988720). Sequence comparisons performed using tBLASTx (30% minimal identity with 10-bp minimum alignment) with percent identity shown as red-shaded regions (inset scale bar). Synteny was recognized when genomes featured a minimum of five consecutive syntenic genes within the same genomic area and separated by a maximum of four nonsyntenic genes.

presented almost the same average distance from a putative V22 common ancestor (1.1456 for *Vequintavirinae* phages and 1.1458 for *Ounavirinae*), the number of syntenic genes present was slightly higher for *Ounavirinae* phages, with 20 syntenic genes on average for *Ounavirinae* versus 15 for *Vequintavirinae* (Fig. 3).

**The V22 structural gene cassette resembles different phages infecting enterobacteria.** Among the predicted structural genes of V22, we identified a cluster of five putative baseplate proteins (gp19 to gp23) and a cluster of three tail-related proteins (gp24, gp26, and gp27). Whole-genome comparison across the phylogenetic tree identified a high degree of synteny within this structural gene product-encoding region (Fig. 4) with all the phages from the *Ounavirinae* clade, as well as most phages from the *Vequintavirinae* clade and an assortment of marine phages such as *Pseudoalteromonas* phage J2-1, MAG GOV_bin_4162, and *Vibrio* phage RYC. As expected, the phage with the highest synteny to V22 across this region was GOV_bin_2917 (Fig. 2B). Otherwise, phages with a similar genome size as V22, such as the *Ounavirinae* phages, presented the closest gene order over the structural region (Table S3). Although a high degree of synteny is observed in the structural region of all the genomes shown in Fig. 4 from the terminase to the baseplate-related gp22, there is a clear drop in synteny from the tail-related proteins (dark blue) onward, most likely due to high genetic variability among these host recognition proteins (52). In the case of *Pseudoalteromonas* phage J2-1 and *Klebsiella* phage vB_kpnM_BIS47, the gene order was less conserved compared to *Ounavirinae* phages.

**Functional gp26 RBP is dependent on the coexpression of gp27.** Adsorption to a suitable bacterial host is the first stage of all phage infections and therefore a critical determinant of phage host range (26). Besides the archetypal T-even phage family,

there are only a few RBPs that have been characterized for marine phages, for instance, gp17 from the T7-like *Prochlorococcus* phage P-SSP7 (53) or gp19 from *Pseudoaltero-monas* phage TW1 (8). As such, by identifying and characterizing the RBP of phage V22, we hoped to expand the repertoire of known phage infection apparatuses for other *Alteromonas* myoviruses or alternative marine phages identified in the future. A combination of structure prediction using the HHpred server (54), BLASTp analysis at the NCBI website, and protein domain identification using the Pfam database (55) were used to identify three proteins that could function as the V22 RBP: gp23, gp24, and gp26. High sequence similarity was observed for gp23 with various phage structural proteins, including lactococcal phage RBPs, and its C terminus was predicted to structurally resemble the receptor-binding distal tips of both the LTF (gp37; 97% probability) (38) and STF (gp12; 97% probability) (56) of phage T4, suggesting a similar receptor-binding functionality. gp24 was also suggested to resemble different phage tail-related proteins; for instance, gp24 has 56% sequence similarity to the minor tail protein (gp177) of *Arthrobacter* phage Racecar (GenBank accession no. MN234206.1) and additionally features structural similarity within its C terminus (Arg900 to Asn1081) to different carbohydrate-binding module domains.

While HHpred did not predict similarity between gp26 and any known structures, a 70-residue segment (Ser193 to Asn263) was predicted to resemble parts of the proximal T4 long tail fiber (gp34) (39) and a *Pseudomonas aeruginosa* R-type pyocin fiber (57), suggesting gp26 could form a similar trimeric beta-helical tail fiber architecture. Furthermore, BLASTp analysis found similarity to phage tail proteins annotated in enterobacterial and *Escherichia coli* genomes, most likely from prophages, which also suggested gp26 could function as a tail fiber. Phages use intra- and intermolecular chaperones (or tail fiber assembly proteins) to ensure correct trimerization and maturation of tail fiber RBPs (58), which in the case of phages T4 and Mu (37, 38) as well as R-type pyocins (57, 59), are encoded directly downstream of the tail fiber genes. While *in silico* analyses could not provide a potential function for gp27, the predicted similarity of gp26 to phage and pyocin fibers combined with the similar size of downstream gp27 (21.8 kDa) to chaperones gp38 (22.3 kDa) and Tfa (20.3 kDa) of phages T4 and Mu, respectively, led us to also test if gp27 functioned as an intermolecular chaperone for gp26.

Fluorescence microscopy was used to assess the ability of the three proteins (green fluorescent protein [GFP]-gp23, GFP-gp24, and GFP-gp26) and GFP-gp26 coexpressed with native gp27 (GFP-gp26_27) to decorate the V22 host, *A. mediterranea* PT11. All constructs were successfully expressed and purified by nickel-nitrilotriacetic acid (Ni-NTA) purification (Fig. 5D; see also Fig. S3). As a side product, His-tagged GFP alone was present in all protein purifications, likely as a side product of overexpression as observed previously (60); however, as shown below, this did not affect our investigations. While none of the individual proteins (including GFP-gp26 alone) demonstrated cell binding (Fig. 5A and C), to our surprise we observed complete cell wall decoration for GFP-gp26 coexpressed with gp27 (Fig. 5A). The host range of the GFP-gp26_27 construct was then tested against other *Alteromonas* spp. using fluorescence microscopy and spectrometry and was shown to additionally bind to a noninfected *Altero-monas macleodii* strain AD45 (Fig. 6A and B). Using phage pulldown assays, we confirmed that whole V22 phage particles could also bind to strain AD45 (60.6% phage adsorption) despite the strain being resistant to V22 infection (Fig. 6C). We further quantified the relative levels of GFP-gp26_27 and GFP-gp26 that bound to strains PT11, AD45, and Te101 (used as a negative control) (Fig. 5B). Whole-cell decoration was observed for GFP-gp26_27 against PT11 with a smaller relative amount of cell decoration observed against the AD45 strain (51%). Again, no binding could be observed for GFP-gp26 produced alone against any of the strains tested, with similar background levels of fluorescence produced (10% versus PT11, 17% versus AD45, and 11% versus Te101) as was also observed for GFP-gp26_27 against the Te101 negative control (21%).

The fluorescence binding assays clearly showed the dependence of GFP-gp26 on coexpression of gp27 to ensure receptor binding functionality. However, the precise

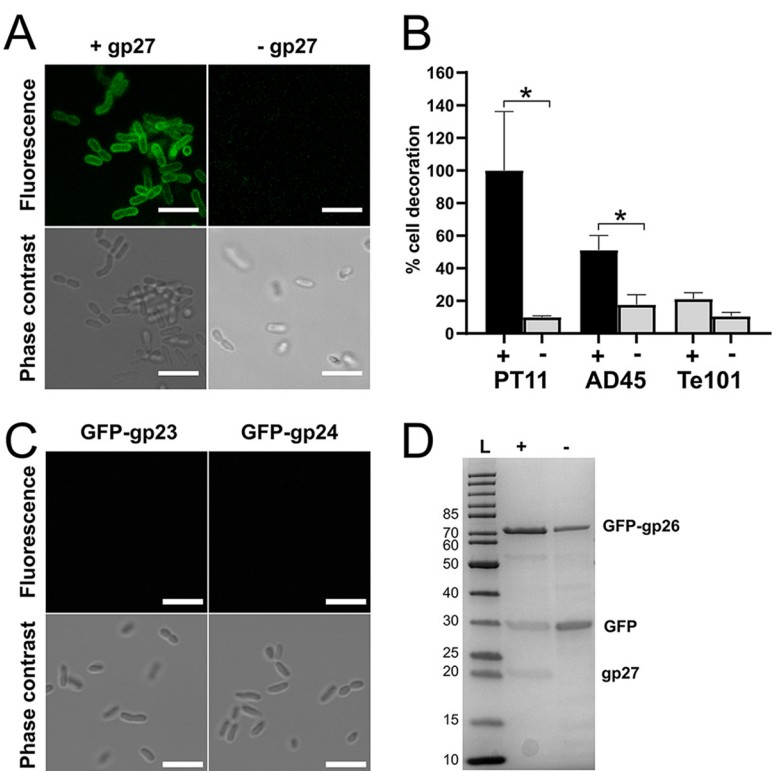

**FIG 5** gp26 is the RBP of phage V22 and requires coexpression of gp27 to be functional. (A) Confocal fluorescence microscopy shows that GFP-gp26 binds to *A. mediterranea* PT11 cells only after coexpression with gp27. Bar, 5 μm. (B) Fluorescence spectrometry measurements of GFP-gp26 interacting with strains PT11, AD45, and Te101 after coexpression with gp27 (+) or alone (−). Percent cell decoration is shown relative to the fluorescence intensity observed for GFP-gp26 + gp27 versus PT11 as the mean from triplicates ± SD. *, $P \leq 0.05$. (C) Confocal fluorescence microscopy revealed no binding for GFP-gp23 or GFP-gp24 to *A. mediterranea* PT11. Bar, 5 μm. (D) SDS-PAGE of Ni-NTA-purified GFP-gp26 (theoretical mass, 68.4 kDa) after coexpression with gp27 (+) or alone (−). A His-tagged GFP contaminant (mass spectrometry-determined mass, 30.2 kDa) always coeluted with both products. A residual amount of gp27 (theoretical mass, 21.8 kDa) was also present after coexpression. L, ladder; numbers at left are molecular masses in kilodaltons.

role of gp27 remained unknown. For instance, SDS-PAGE analysis revealed a residual amount of gp27 always present after elution of GFP-gp26 from the Ni-NTA resin (Fig. 5D). We believed this could be due to two possible reasons: (i) gp27 functions as an intermolecular chaperone, similar to the gp38 of the T4 LTF (38), and its coelution is simply due to nonspecific binding of gp27 to the Ni-NTA resin, or (ii) gp27 can incorporate into the mature GFP-gp26 RBP complex, similarly to the terminally attached proteins of Mu, S16, and other T2-like tail fibers (36, 37, 41), which dissociate during SDS-PAGE analysis.

Using size-exclusion chromatography (SEC) and assessing individual peak composition by SDS-PAGE, we tested if GFP-gp26 and gp27 eluted together in a single peak (which would suggest an interaction) or separately. SEC produced a three-peak pattern with GFP-gp26 eluting as a single peak (8 to 10 ml) followed by individual peaks of gp27 (12 ml) and the GFP side product (13 ml) (Fig. 7A, blue line). None of the residual gp27 could be identified by SDS-PAGE within the first GFP-gp26 peak, suggesting no strong interaction exists between gp26 and gp27 (Fig. 7C). By comparison, the fiber chaperone (Tfa) of the Mu tail fiber remained bound to the distal tip of the fiber when tested using a similar combination of Ni-NTA and SEC purification (37). Nevertheless, the interaction between gp26 and gp27 could be driven by hydrophobic interactions and be affected by salt concentration (61). We therefore performed the same analysis using a high-salt SEC buffer (25 mM Tris, 500 mM NaCl, pH 7.4) and analyzed peak composition. A similar peak pattern was again produced with GFP-gp26 eluting first as

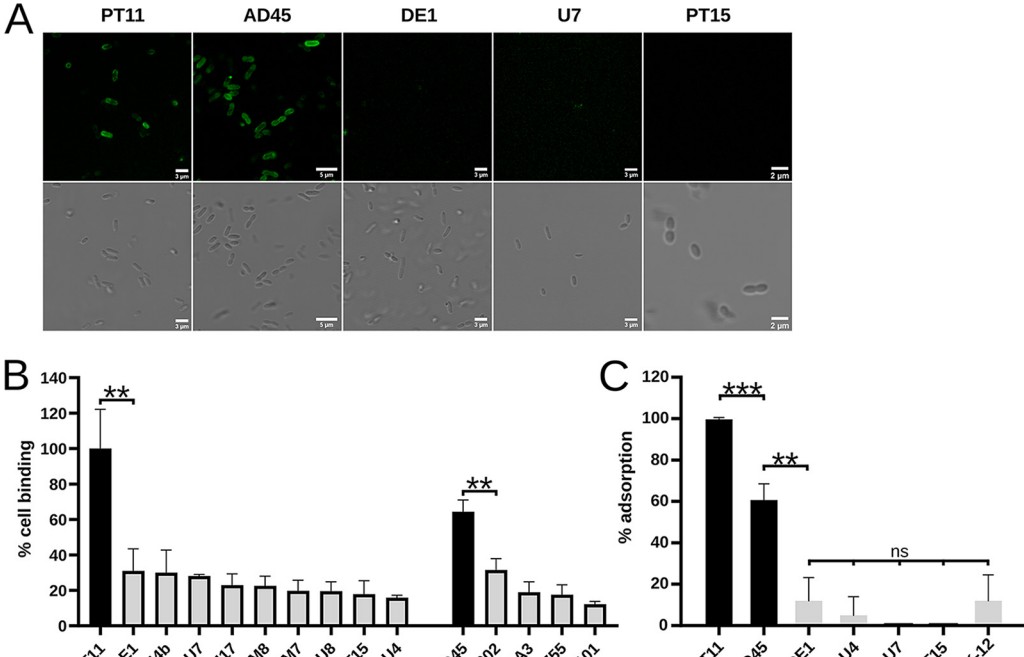

**FIG 6** Host binding range of GFP-gp26 RBP and whole V22 phages. (A and B) Fluorescence microscopy (A) and spectrometry (B) determination of the binding of GFP-gp26 (coexpressed with gp27) against different *A. mediterranea* and *A. macleodii* strains. Binding was observed only for *A. mediterranea* strain PT11 and *A. macleodii* strain AD45. Percent cell binding is determined relative to strain PT11. (C) V22 phage adsorption was observed only to *Alteromonas* strains PT11 and AD45. Percentage of adsorbed phages was determined relative to the number of unbound phages [formula: (total phages − unbound phages)/total phages; see Materials and Methods]. Percent cell binding by GFP-gp26 (B) or phage particles (C) calculated as the mean from triplicates ± SD. ***, $P \leq 0.001$. **, $P \leq 0.01$. *, $P \leq 0.05$; ns, not significant ($P > 0.05$).

two overlapping peaks followed by peaks of gp27 and the GFP side product (Fig. S4). In addition, we performed the same SEC analysis using GFP-gp26 expressed in the absence of gp27 (Fig. 7A, orange line) and analyzed peak composition (Fig. 7C). Regardless of the salt concentration (0 mM or 500 mM) of the running buffer, two separated peaks for GFP-gp26 and the GFP side product were produced with similar retention volumes as observed for GFP-gp26 coexpressed with gp27.

We next tested if GFP-gp26 purified from any residual gp27 was still capable of interacting with PT11 host cells. Individual fractions (peaks 1) of purified GFP-gp26 after gp27 coexpression or produced alone were collected, and their ability to decorate *A. mediterranea* PT11 cells was assessed using fluorescence microscopy (Fig. 7B). The purified GFP side product was combined and used as a negative control. As expected, GFP and purified GFP-gp26 without gp27 coexpression did not bind to PT11 cells. Meanwhile, purified GFP-gp26 after gp27 coexpression remained functional and bound strongly to PT11 cells. This demonstrated that postexpression, GFP-gp26 no longer requires gp27 for host recognition.

Overall, we present evidence that gp26 and gp27 do not interact with each other postexpression and, furthermore, that mature gp26 does not require the presence of gp27 to bind bacterial cells. In conclusion, we suggest gp27 represents a new type of intermolecular chaperone that is essential for expression and folding of functional V22 RBP.

## DISCUSSION

The marine phage V22 represents a new putative genus within the *Myoviridae* family, here named *Myoalterovirus* (under ICTV review). Whole-genome analysis revealed no sequence similarity between V22 and other known phages except for an uncultured marine phage assembled from a viral metagenome named GOV_bin_2917.

 

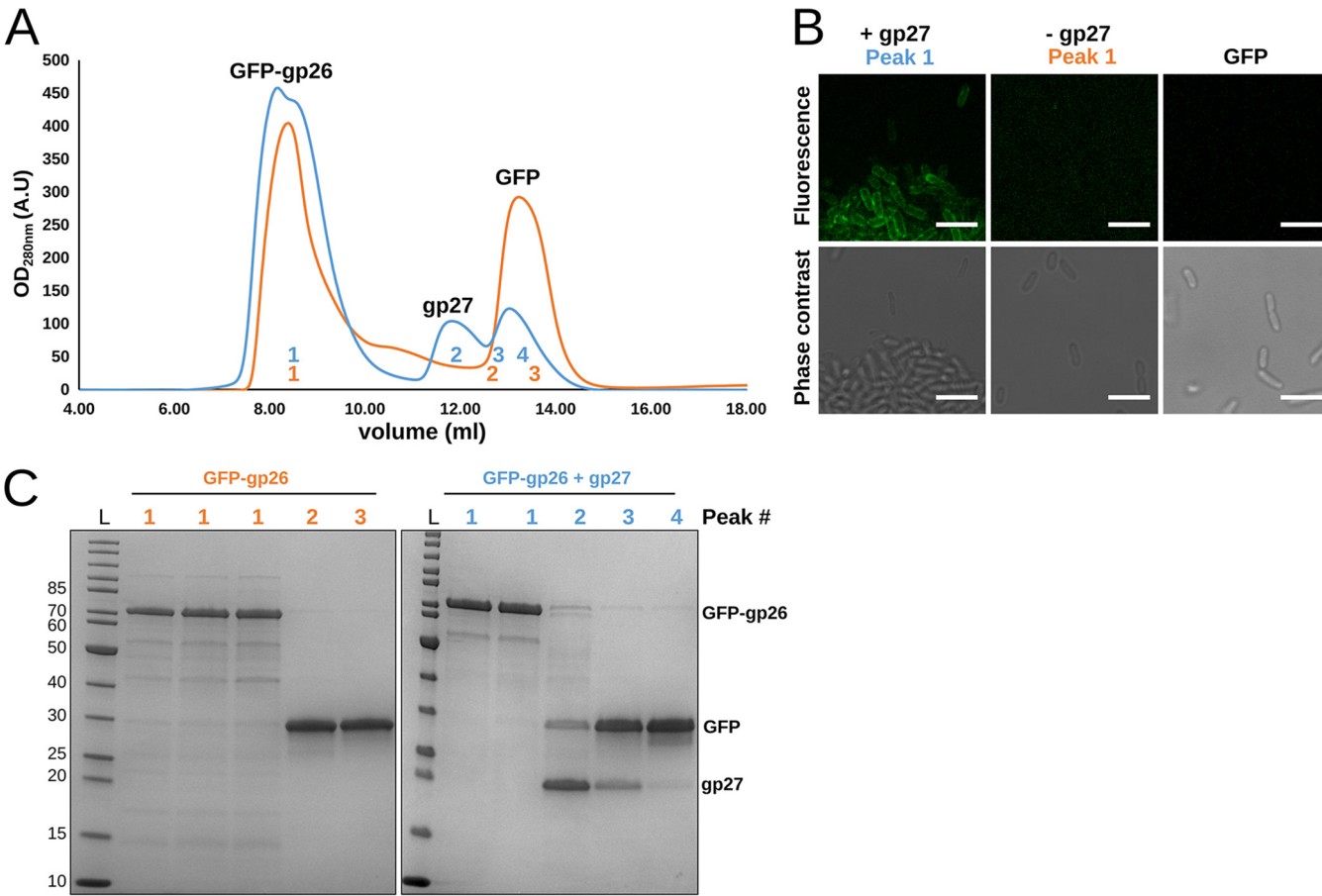

**FIG 7** gp26 retains its receptor-binding function after SEC-based separation from the gp27 intermolecular chaperone. (A) Size-exclusion chromatography (SEC) elution profile (UV trace at 280 nm) of GFP-gp26 after coexpression with gp27 (blue) or alone (orange). In both cases, GFP-gp26 always eluted as a single peak (8 to 10 ml) followed by peaks for gp27 alone (12 ml) and the GFP side product (13 to 14 ml). (B) Fluorescence microscopy of the SEC-purified GFP-gp26 after gp27 coexpression (blue peak 1), GFP-gp26 produced alone (orange peak 1), and the GFP contaminant to *A. mediterranea* PT11. Bar, 5 μm. (C) SDS-PAGE analysis of individual peaks produced by SEC shows complete separation of gp27 and the GFP side product from GFP-gp26 expressed with or without gp27. L, protein ladder.

Furthermore, V22 and GOV_bin_2917 both present significant phylogenetic distance to their closest relatives (Fig. 3), providing further support for their classification as a new genus. Despite low overall sequence similarity with other phages, we identified strong conservation of gene synteny within the structural region of V22 with various phages infecting enterobacteria (Fig. 4). It is a common observation to find some degree of homology or synteny between genes encoding viral structural proteins when different related tailed-bacteriophage genomes are compared (62–66). Nevertheless, the high level of synteny, yet lack of sequence similarity, within the baseplate genes of marine phages and phages targeting enterobacteria provides a clear example of how essential viral functions, i.e., host adsorption and DNA ejection through the baseplate, have evolved to provide the most effective gene arrangements despite the continuous adaptation of these phages in substantially different environments. Unexpectedly, we found more synteny and homology between V22 and the phages infecting enterobacteria (*Ounavirinae*) than between V22 and other marine phages analyzed in this study; however, the minimal number of marine myovirus genomes available could bias this result. Nevertheless, the fact that phages similar to V22 were identified, such as GOV_bin_2917 retrieved from metagenomes of the Indian Ocean, or the *Rheinheimera* phage vB_RspM_Barba19A isolated from the Baltic Sea, suggests a widespread presence of *Myoalterovirus*-like phages in the oceans.

The diversity of the tail-related genes across the V22 syntenic phages is remarkable.

Indeed, a high level of variability among tail fibers has been reported several times before (67–72), and as expected, the tail fiber, gp26, presented the highest level of variation across the structural regions of both myoalteroviruses. Interestingly, sequence variability was highest within the C-terminal region of gp26 and its GOV_bin_2917 homolog, including an insertion within the central region of the latter (see Fig. S5A in the supplemental material). RBPs typically contain receptor-binding sites within the C terminus (24), and as such, the decrease in C-terminal conservation between the two RBPs could be due to variability in binding mechanisms to recognize their different hosts.

During host range analysis with different *Alteromonas* strains, we observed only V22 infection (determined by plaque formation) against its propagation host, *A. mediterranea* PT11. However, phage pulldown assays revealed that the phage and its RBP (gp26) could also interact with the *A. macleodii* strain AD45 (Fig. 6C). No lysogeny-related genes (e.g., integrase) were identified within the V22 genome, suggesting that the lack of visible infection in AD45 is not due to lysogenic conversion. Furthermore, no CRISPR-Cas system or prophages could be identified within the genome of AD45, suggesting the lack of infectivity is also not due to bacterial immunity or prophage-induced immunity. While other antiphage defenses such as abortive infection or restriction modification systems could be responsible for terminating a potential V22 infection, it would be of great interest to find V22-like phages that do infect strain AD45 to then test if certain genetic elements, e.g., the AMGs carried by V22 and other marine phages (48), play critical roles in determining a successful phage infection.

Here, we show that gp26 RBP is capable of host cell recognition only when coexpressed with gp27, which was thus subsequently identified as an intermolecular chaperone. Various types of phage chaperones have been shown previously to assist with maturation and formation of phage tailspike and tail fiber RBPs. For instance, the tailspikes of *E. coli* K1F-specific phages (73, 74) and the *Bacillus subtilis*-targeting phage φ29 (75), as well as tail fibers of *Salmonella* phage S16 (41) and *E. coli* phage T5 (76), all feature C-terminal intramolecular chaperone domains that assist with trimerization before autoproteolysis removes them from the mature RBP complex (77). Similarly, tail fibers of R-type pyocins (57, 59) and phage T4 (58) use intermolecular chaperones encoded immediately downstream of their tail fiber genes that assist in trimerization beginning at the C-terminal end of the fiber but do not incorporate into the final complex. Interestingly, no homology could be identified between the V22 gp27 and any of the known phage chaperones, with BLASTp analysis identifying similarity only to the GOV_bin-2917 homolog (gp50) and various hypothetical proteins, all of which contained a C-terminal domain of unknown function (DUF4376). All of this led us to identify V22 gp27 as a new type of intermolecular chaperone for V22-like RBP maturation.

We are currently investigating the host receptor of V22 that is present in *A. macleodii* and *A. mediterranea* strains. We are also investigating the coevolution of phage V22 with its propagation strain to analyze genomic variation over time, and thanks to the experimental results obtained with the RBP in this study, we will be able to analyze with more confidence the possible genetic variations found in this genetic structure and its potential evolutionary implications. Future analyses of newly isolated *Alteromonas* phages will increase our knowledge about the genomic diversity present in these viruses and could shed light on new and interesting evolutionary aspects related with these marine tailed bacteriophages and their associated recognition structures.

## MATERIALS AND METHODS

**Bacterial strains.** All *Alteromonas* strains used in this study are listed in Table 1. *Alteromonas* strains were grown in marine medium (MM) (sea salts 3.5% [Sigma], yeast extract 0.1% [Scharlau], and peptone 0.5% [PanReac-AppliChem]) at room temperature under agitation. *E. coli* XL1-Blue MRF′ cells (Stratagene) were used for all cloning steps, plasmid transformations, and protein expression. *E. coli* K-12 was used as a negative-control strain for phage pulldowns and GFP binding assays. Both were grown in LB medium at 37°C with agitation.

**Phage isolation.** Phage name assignation, *Alteromonas* virus vB_AmeM_PT11-V22 (V22 for short), was formulated according to the nomenclature described by Kropinski et al. (78). V22 was isolated from western Mediterranean coastal waters, at Villajoyosa, Alicante (Spain), in July 2016. The V22 propagation host, *Alteromonas mediterranea* PT11 (GenBank accession no. NZ_CP041169), was isolated at Postiguet beach, Alicante, approximately 35 km away from Villajoyosa, during the same 2016 sampling campaign. Ten milliliters of filtered seawater (0.22-$\mu$m cellulose acetate membrane filters [Sartorius]) was mixed with 15 ml of marine medium, spiked with 100 $\mu$l of an overnight *A. mediterranea* PT11 culture, and incubated at 25°C for 24 h with agitation. The enrichment was centrifuged (10,000 $\times$ *g*, 4°C, 15 min), and the supernatant was collected and filtered again (0.22 $\mu$m) before phage isolation using the double agar overlay technique (79). In brief, the enriched phage solution was diluted 100-fold and 100 $\mu$l was mixed with warmed 3-ml 0.7% marine agar spiked with 100 $\mu$l of an overnight *A. mediterranea* PT11 culture, which was subsequently poured onto an agar plate containing 1.5% marine agar and incubated at 25°C for 16 h. Individual plaques were picked and resuspended in 100 $\mu$l SM buffer (50 mM Tris,100 mM NaCl, 10 mM magnesium sulfate, pH 7.5), and the double agar overlay plaque purification was repeated two more times prior to final phage stock preparation using both solid and liquid media as described by Gutiérrez et al. (80) with the final phage solution filtered using 0.22-$\mu$m membrane filters. The final PFU/ml concentration was determined using the double agar overlay technique, and long-term phage stocks were established at 4°C and −80°C (with 20% glycerol added).

**DNA extraction.** A 12.5 mM concentration of MgCl$_2$, 5 U of DNase I (Thermo Fisher Scientific), and 0.3 mg/ml RNase A (Thermo Fisher Scientific) were added to 1 ml of phage stock (4.7 $\times$ 10$^{10}$ PFU/ml) and incubated at room temperature for 30 min. A 20 mM concentration of EDTA, 0.05 mg/ml proteinase K (PanReac-AppliChem), and 0.5% SDS were then added, and the mixture was incubated at 55°C for 60 min. Phage DNA was extracted using the phenol-chloroform method with MaXtract high-density Eppendorf tubes (Qiagen) and precipitated by adding 0.1 volumes 3 M sodium acetate and 2.5 volumes of 100% ethanol. The precipitated DNA was resuspended in 50 $\mu$l of water, and concentration was determined using a Qubit 3.0 fluorometer (Invitrogen) before storage at −20°C.

**Assembly, annotation, and genome analysis.** The genome was sequenced using an Illumina MiSeq instrument (2 $\times$ 300bp) at FISABIO facilities (Valencia, Spain). Sequenced reads were quality checked using FastQC v0.11.7 (Babraham Bioinformatics) and quality trimmed using Trimmomatic v0.32 (81). Assembly of the reads was carried out using CLC Genomics Workbench v8.0 (Qiagen Bioinformatics) with respective default options suggested for genome assemblies, and the process yielded one single contig. A coverage quality analysis of the reads was performed by aligning genomic reads to the assembled contig using Bowtie 2 v1.3.1 (82) and SAMtools v1.2 (83). Results were visually checked using Tablet (84).

Coding DNA sequences (CDSs) were predicted using Prodigal (85), and genes were annotated using Diamond v0.9.4.105 (86) against the nonredundant NCBI database (E value was set at 1e−3) and hmmscan (HMMER v3.1b2) (87) against pVOGs and Pfam databases (55, 88) (E value <1e−3). Possible inconsistencies eventually produced by different annotation tools were checked manually using the InterPro database (89), the Conserved Domain Database suite (CDD/SPARCLE) (90), the UniProtKB database (91), and the HHpred server (54). Easyfig v2.2.2 (92) was used for genome visualization. For tRNA gene detection, ARAGORN (93) and tRNAscan-SE (94) were used. CMsearch (95) and StructRNAfinder (96) were used for screening the presence of noncoding RNA (ncRNA). The Rfam database (97) was employed for both tRNA and ncRNA searching. CRISPR searching in *Alteromonas* strains PT11 and AD45 was performed using CRISPRCasFinder (98) and CRISPI (99). Prophage searching in the same strains was performed using VirSorter (100) and PHASTER (101). Annotation for the MAG GOV_bin_2917 was performed in the same way as for the V22 genome. *A. mediterranea* PT11 and *A. macleodii* AD45 CDSs were predicted and annotated using Prodigal and Diamond (against the nonredundant NCBI database), respectively.

In order to study the genetic variability of gp26 and gp27 compared to other similar known sequences existing in the databases, BLAST alignments were performed using the UniProtKB database with an E value of <1e−3 and percent identity of ≥30, and obtained sequences were downloaded for further analysis. The CD-HIT suite (102) with default parameters and 100% identity cutoff was implemented to remove sequence redundancy. In total, 115 gp26 similar sequences and 33 gp27 similar sequences were selected and MUSCLE aligned (103). Also, BLASTp analysis (E value <5e−3) was performed with all the selected sequences.

**Phylogenetics.** A database of 1,220 large subunit terminase (TerL) amino acid sequences created by Mizuno et al. (104) was used to analyze the phylogenetic relationship between V22 and other *Caudovirales*. An additional 65 terminase genes with 46% to 84% sequence similarity to the V22 terminase were identified by BLASTp from the nonredundant database and added to our database, providing a total of 1,285 sequences. Sequences were aligned using MUSCLE (103,) and a first phylogenetic tree was generated using FastTreeMP v2.1.7 (105). The obtained phylogenetic tree allowed the selection of 42 sequences within the same monophyletic group to the V22 terminase. Terminases from T4 and seven other T4-like phages were added as an outgroup. The final data set with 50 sequences was MUSCLE aligned, and a more accurate phylogenetic tree was computed using IQ-TREE v1.6.11 (106). The selection of the substitution model for tree construction was chosen using ModelFinder (107), which determined LG and G4 models as the most optimal. The final phylogenetic tree was generated using maximum likelihood, and 1,000 bootstrap replications with final topologies were visualized using iTOL v4 (108).

**Genome comparative analysis.** Fifty-six phage genomes were used for comparative analysis to identify homology and synteny to phage V22. They were selected after BLASTp analysis with the V22 terminase gene and other annotated genes against the nonredundant database based on the high homology detected. Only genomes holding genes with similarity values between 45% and 85% were

selected for analysis. In addition, 10 representative *Myoviridae* genomes (i.e., T4, Mu, P2, and seven T4-like phages) were included in the analysis (Tables S3 and S4). Sequence comparisons were performed using tBLASTx and 30% minimal identity on 10-bp minimum alignments. In Fig. 4, V22 and seven syntenic structural regions selected from the previous analysis were aligned and compared. Synteny was considered when the genomic region showed at least five or more syntenic genes in a row (i.e., similar size and gene order) with a maximum separation of four nonsyntenic genes.

**Phage host range.** The host range of phage V22 was tested against 15 *Alteromonas* species strains (Table 1) using a spot test infection assay. Briefly, 10 $\mu$l of a phage V22 serial dilution in SM buffer ($10^{10}$ to $10^5$ PFU/ml) or 10 $\mu$l SM buffer alone (control) was spotted onto a bacterial lawn containing 200 $\mu$l of an overnight bacterial culture mixed with 3 ml marine soft agar and incubated at 25°C for 16 h. Formation of individual plaques provided confirmation of successful host infection.

**Transmission electron microscopy.** Phage V22 was concentrated and purified by CsCl isopycnic centrifugation and dialyzed into SM buffer to reach $\sim10^{11}$ PFU/ml. Phage particles were added to carbon-coated copper grids (Quantifoil) and negatively stained for 20 s using 2% uranyl acetate. Grids were observed at 100 kV using a Hitachi HT 7700 scope equipped with an AMT XR81B Peltier cooled charge-coupled device (CCD) camera (8M pixel).

**GFP-tagged RBP construct generation.** Selected gene fragments (*gp23*, *gp24*, *gp26*, and *gp26_gp27*) and a fragment of the expression plasmid backbone were inserted into a pQE30 derivative plasmid, pQE30_HGT (60), featuring an N-terminal His tag connected to GFP via a tobacco etch virus (TEV) cleavage site next to which RBP gene fragments were introduced individually using Gibson assembly (New England BioLabs). Gene fragments were generated by PCR with primers listed in Table S5 and using V22 phage particles and pQE30_HGT as the templates. All cloned plasmids were transformed into *E. coli* XL1 Blue MRF′ cells, purified, and Sanger sequenced to ensure correct insertions.

**Protein expression and purification.** An overnight culture of plasmid-transformed *E. coli* XL1 Blue MRF′ cells was used to inoculate 500 ml LB medium supplemented with 100 $\mu$g/ml ampicillin and 15 $\mu$g/ml tetracycline and grown with agitation at 37°C until reaching an optical density at 600 nm (OD$_{600}$) of 0.6. The cultures were cooled to 20°C, induced with 0.5 mM isopropyl-$\beta$-D-thiogal-actopyranoside (IPTG), and incubated for 16 h with agitation at 20°C. Cells were harvested by centrifugation at 5,500 $\times$ *g* for 15 min, resuspended in 30 ml of buffer A (50 mM Na$_2$HPO$_4$, pH 8.0, 500 mM NaCl, 5 mM imidazole, 0.1% Tween 20) at 4°C, and lysed using a Stansted pressure cell homogenizer (Stansted Fluid Power). The cell extract was centrifuged to remove unbroken cells at 15,000 $\times$ *g* for 60 min prior to immobilized-metal affinity chromatography (IMAC) using low-density Ni-nitrilotriacetic acid (NTA) resin (Agarose Bead Technologies). Buffer A was used to wash beads before eluting proteins using buffer A plus 250 mM imidazole. The proteins were subsequently dialyzed for 16 h into 25 mM Tris, pH 7.5, and stored at 4°C. Purified proteins were assessed using SDS-PAGE. Twenty micrograms of dialyzed protein was mixed with Laemmli sample buffer (Bio-Rad), treated with or without heat denaturation (96°C, 8 min), and run on a TGX stain-free precast gel (Bio-Rad) for 38 min at 200 V. Protein bands were visualized using both UV absorbance (280 nm) and InstantBlue Coomassie staining (Expedeon) on a Gel Doc XR+ imaging system (Bio-Rad).

**Fluorescence microscopy and spectrometry of GFP-tagged proteins.** Five hundred microliters of an overnight bacterial culture at an OD$_{600}$ of 0.5 was collected by centrifugation (6,000 $\times$ *g*, 5 min) and resuspended in fresh MM. Eighty micrograms of GFP-tagged protein was added to the cells and mixed using an overhead rotator for 30 min at room temperature. Cells were collected by centrifugation (6,000 $\times$ *g*, 5 min), and the supernatant containing unbound protein was discarded. The cell pellet was then washed by resuspension in 1 ml of MM followed by centrifugation. Washing was repeated two more times before final resuspension of the pellet in 200 $\mu$l of MM. One hundred fifty microliters was added to individual wells of a black, flat-bottom microplate (Greiner Bio-One), and fluorescence intensity of bound GFP-tagged proteins was measured at ambient temperature using a POLARstar Omega spectro-photometer (BMG Labtech) at 485-nm excitation, 520-nm emission with (1,000$\times$) fixed gain. The reported percent cell decoration is relative fluorescence intensity compared to *A. mediterranea* PT11. Statistical analyses and graphs were generated using Prism 8.0 (GraphPad Software). Triplicate assays are shown as the mean $\pm$ standard deviation. Student's *t* test was performed with a confidence level of 95%. For fluorescence microscopy, 4 $\mu$l of the cell suspension was imaged using a confocal inverted microscope (Leica TCS SPE) equipped with an ACS APO 63$\times$/1.30 oil CS lens objective with excitation at 488 nm and emissions collected with a PMT detector in the detection range of 510 to 550 nm. Transmitted-light microscopy images were obtained with the differential interference contrast mode. Images were acquired with a Leica DFC 365 FX digital camera controlled with the LAS AF software. Fiji v2.0.0 (ImageJ software) was used to produce the final microscopy images.

**GFP-gp26 size-exclusion chromatography.** A 2.5-mg amount of dialyzed GFP-gp26_gp27 (expressed with chaperone) or GFP-gp26 (expressed without chaperone) was analyzed by size-exclusion chromatography using a Superdex 200 10/300 column (GE Life Sciences) in both low-salt (25 mM Tris, pH 7.4) and high-salt (25 mM Tris, 500 mM NaCl, pH 7.4) running buffers on an ÄKTA purifier fast-protein liquid chromatograph (FPLC) (Amersham Biosciences) with 0.5-ml/min flow speed. Peaks were detected at wavelengths of 280 nm, 260 nm, and 315 nm and collected separately using a Frac-950 fraction collector (Amersham Biosciences). Protein content of each peak was analyzed by SDS-PAGE and imaged using UV absorbance and Coomassie blue staining, as described above. Individual peak fractions were combined together and concentrated to a final concentration of 1 mg/ml for fluorescence microscopy as described above.

**Whole-phage particle binding assay.** Overnight cultures of bacteria were adjusted to an OD$_{600}$ of 1.0, corresponding to $\sim10^9$ CFU, in 1 ml of MM reaction volume. A total of $10^7$ PFU of V22 phage was

added to the bacterial cells and mixed on an overhead rotator at room temperature for 10 min, providing sufficient time for phage-host adsorption without phage replication. Bacterium-bound phages were then removed by centrifugation (20,000 $\times$ *g*, 5 min), and the PFU/ml of all nonadsorbed phages remaining in the supernatant was determined using soft agar overlay assays against *A. mediterranea* PT11. The adsorption ratio was determined by comparison to the total amount of phages added, determined from a control reaction of $10^7$ PFU of V22 phage added to 1 ml MM containing no bacteria. Triplicate assays are shown as the mean $\pm$ standard deviation. Prior to statistical analysis, all negative adsorption ratios were normalized to zero. Student's *t* test was performed with a confidence level of 95%. GraphPad and SigmaPlot v11.0 were used for statistical analysis and graphic visualization.

**Data availability.** The annotated genome sequences for vB_AmeM_PT11-V22 and its host *A. mediterranea* PT11 are available from the GenBank database under accession numbers MN877442 and NZ_CP041169, respectively.

## SUPPLEMENTAL MATERIAL

Supplemental material is available online only.

**FIG S1**, TIF file, 0.3 MB.

**FIG S2**, TIF file, 1.7 MB.

**FIG S3**, TIF file, 2.6 MB.

**FIG S4**, TIF file, 1.4 MB.

**FIG S5**, TIF file, 0.9 MB.

**TABLE S1**, DOCX file, 0.01 MB.

**TABLE S2**, DOCX file, 0.02 MB.

**TABLE S3**, DOCX file, 0.02 MB.

**TABLE S4**, DOCX file, 0.01 MB.

**TABLE S5**, DOCX file, 0.01 MB.

## ACKNOWLEDGMENTS

Special thanks to Diana Gutiérrez and Pilar García for assistance and training in lab phage management techniques. We also thank J. Jeffrey Morris, Allison Coe, Shengwei Hou, and Daniel Sher for the shipment of *Alteromonas* strains EZ55, MIT1002, Te101, and HOT1A3, respectively.

This work was supported by grants 'VIREVO' CGL2016–76273–P (MCI/AEI/FEDER, EU) (cofounded with FEDER funds) from the Spanish Ministerio de Ciencia e Innovación and 'HIDRAS3' PROMETEU/2019/009 from Generalitat Valenciana. R.G.-S. was supported by a predoctoral fellowship from the Valencian Consellería de Educació, Investigació, Cultura i Esport (ACIF/2016/050) and was also a beneficiary of the BEFPI 2019 fellowship for predoctoral stays from Generalitat Valenciana and The European Social Fund. F.R.-V. was a beneficiary of the 5top100 program of the Ministry for Science and Education of Russia.

Conceptualization, F.R.-V., R.G.-S., and M.D.; funding acquisition, F.R.-V., R.G.-S., and M.J.L.; investigation, R.G.-S., M.D., V.G., L.V.Z., J.J.R.-G., and A.-B.M.-C.; project administration, F.R.-V. and M.D.; resources, F.R.-V. and M.J.L.; supervision, F.R.-V., M.D., R.R., and M.J.L.; validation, M.D. and R.G.-S.; visualization, R.G.-S. and M.D.; writing— original draft, R.G.-S. and M.D.; writing—review and editing, R.G.-S., M.D., R.R., A.-B.M.-C., and F.R.-V.

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
