## [Reviewer comments · mSystems]

***Alteromonas myovirus V22* represents a new genus of marine bacteriophages requiring a tail fiber chaperone for host recognition**

Rafael Gonzalez-Serrano, Matthew Dunne, Riccardo Rosselli, Ana Martin-Cuadrado, Virginie Grosboillot, Léa Zinsli, Juan Roda-Garcia, Martin Loessner, and Francisco Rodriguez-Valera

Corresponding Author(s): Francisco Rodriguez-Valera, Universidad Miguel Hernandez

Review Timeline:

Submission Date:	March 9, 2020
Editorial Decision:	May 6, 2020
Revision Received:	May 15, 2020
Accepted:	May 20, 2020

Editor: David Knipe

Reviewer(s): Disclosure of reviewer identity is with reference to reviewer comments included in decision letter(s). The following individuals involved in review of your submission have agreed to reveal their identity: Xiaoxue Wang (Reviewer #2)

Transaction Report:

DOI: <https://doi.org/10.1128/mSystems.00217-20>

May 6, 2020

Prof. Francisco Rodriguez-Valera
Universidad Miguel Hernandez
Division de Microbiologia
Campus de San Juan
Apartado 18
San Juan de Alicante, Alicante 03550
Spain

Re: mSystems00217-20 (***Alteromonas Myovirus V22* represents a new genus of marine bacteriophages requiring a tail fiber chaperone for host recognition**)

Dear Prof. Francisco Rodriguez-Valera:

Your manuscript has been reviewed by two reviewers and they have found it interesting and an interesting contribution. They have some requested modifications that will improve the manuscript and I ask that you address them in a revised manuscript. In particular, Reviewer 2 raises some important questions about whether the evidence is really that strong for a new genus. This needs to be addressed in a revised manuscript.

Below you will find the comments of the reviewers.

To submit your modified manuscript, log onto the eJP submission site at <https://msystems.msubmit.net/cgi-bin/main.plex>. If you cannot remember your password, click the "Can't remember your password?" link and follow the instructions on the screen. Go to Author Tasks and click the appropriate manuscript title to begin the resubmission process. The information that you entered when you first submitted the paper will be displayed. Please update the information as necessary. Provide (1) point-by-point responses to the issues raised by the reviewers as file type "Response to Reviewers," not in your cover letter, and (2) a PDF file that indicates the changes from the original submission (by highlighting or underlining the changes) as file type "Marked Up Manuscript - For Review Only."

Due to the SARS-CoV-2 pandemic, our typical 60 day deadline for revisions will not be applied. I hope that you will be able to submit a revised manuscript soon, but want to reassure you that the journal will be flexible in terms of timing, particularly if experimental revisions are needed. When you are ready to resubmit, please know that our staff and Editors are working remotely and handling submissions without delay. If you do not wish to modify the manuscript and prefer to submit it to another journal, please notify me of your decision immediately so that the manuscript may be formally withdrawn from consideration by mSystems.

To avoid unnecessary delay in publication should your modified manuscript be accepted, it is important that all elements you upload meet the technical requirements for production. I strongly recommend that you check your digital images using the Rapid Inspector tool at <http://rapidinspector.cadmus.com/RapidInspector/zmw/>.

If your manuscript is accepted for publication, you will be contacted separately about payment

when the proofs are issued; please follow the instructions in that e-mail. Arrangements for payment must be made before your article is published. For a complete list of **Publication Fees**, including supplemental material costs, please visit our website.

Sincerely,

David Knipe

Editor, mSystems

Journals Department
Reviewer comments:

Reviewer #1 (Comments for the Author):

A truly excellent manuscript describing the full characterization of an *Alteromonas* phage, with particular reference to its receptor-binding proteins.

Line 1 (and elsewhere) - change "Myovirus" to "myovirus"

Line 43 - call me old fashioned by *Enterobacteria* is a legitimate word (see <https://www.ncbi.nlm.nih.gov/pubmed/29785663>) has no taxonomic rank, and should never be italicized. It is equivalent to *pseudomonad* or *coliform*.

Line 81 - change "Members of *Alteromonas* populate..." to "Members of genus *Alteromonas* populate..."

Fig. 1 - excellent quality (Hans Ackermann would have been delighted)

Line 232 - change "*Myoalteroviridae*" to "*Alteromonas myoviruses*" The former term refers to a family of related phages, which you have not proposed.

Reviewer #2 (Comments for the Author):

The manuscript by Gonzalez-Serrano et al. identified a lytic phage infecting *Alteromonas* and classified it into a new genus *Myoalterovirus*. Furthermore, they found that tail-related proteins in the structural region displayed a high genetic variability. Through fluorescence microscopy and spectrometry, they concluded that gp26 was the RBP and confirmed the dependency on the co-expression of gp27. Gp27 was identified as an intermolecular chaperone. The interplay of phage

and host is an important topic. Overall, the experiments and analysis is conducted well and the manuscript was written clearly in general. However, the evidence that support that V22 represents a new genus is mainly phage genomic analysis. Other features such as the morphology and RBPs of this phage resemble T4 phage.

Major and minor concerns:

1. On lines 270, Results showed that GFP-gp26/gp27 can bind to a non-infected *A. macleodii* strain AD45 (Figure 6A) and the same for the phage absorption test (Figure 6B). The authors argued that since there is no CRIPSR or prophage in the genome of AD45, thus it is not due to host immunity. This is not true since it is possible that AD45 has other types of anti-phage defense mechanisms such as RM systems or abortive infection systems.

2. There are some concerns of the GFP contamination in the co-expression assay of Gp27 and Gp26

In Figure 5, His-tagged GFP contaminants were both presented in GFP-gp26 and GFP-gp26/gp27, but the GFP protein varied quite a lot in two conditions.

On Line 532 "amounts indicated in figure legends".

These amounts are not included in Figures and Figure legends.

Dear Professor Knipe,

We would like to thank you and the reviewers for your time invested in evaluating our work. We are sincerely grateful to both reviewers for providing informative reviews, from which we have taken on board all comments raised to improve the quality of the paper.

Please find below our responses to the reviewers' comments:

Reviewer #1 (Comments for the Author):

A truly excellent manuscript describing the full characterization of an *Alteromonas* phage, with particular reference to its receptor-binding proteins.

After dedicating a significant amount of time and effort into understanding V22 and describing this new genus, we are highly grateful for this response from the reviewer.

Line 1 (and elsewhere) - change "Myovirus" to "myovirus"

Done.

Line 43 - call me old fashioned by Enterobacteria is a legitimate word (see <https://www.ncbi.nlm.nih.gov/pubmed/29785663>) has no taxonomic rank, and should never be italicized. It is equivalent to pseudomonad or coliform.

Thank you for this information. We have now corrected this throughout the manuscript, including on line 255 and also within the Supplementary table S4, where genus was not properly assigned for the first 3 viruses.

Line 81 - change "Members of *Alteromonas* populate..." to "Members of genus *Alteromonas* populate..."

Done.

Fig. 1 - excellent quality (Hans Ackermann would have been delighted)

Thank you very much for this kind comment.

Line 237 - change "*Myoalteroviridae*" to "*Alteromonas* myoviruses" The former term refers to a family of related phages, which you have not proposed.

Done.

Reviewer #2 (Comments for the Author)

The manuscript by Gonzalez-Serrano et al. identified a lytic phage infecting *Alteromonas* and classified it into a new genus *Myoalterovirus*. Furthermore, they found that tail-related proteins in the structural region displayed a high genetic variability. Through fluorescence microscopy and spectrometry, they concluded that gp26 was the RBP and confirmed the dependency on the co-expression of gp27. Gp27 was identified as an intermolecular chaperone. The interplay of phage and host is an important topic. Overall, the experiments and analysis is conducted well

and the manuscript was written clearly in general. However, the evidence that support that V22 represents a new genus is mainly phage genomic analysis. Other features such as the morphology and RBPs of this phage resemble T4 phage.

Thank you for your kind review. We understand that our explanation behind the classification of V22 and GOV_bin_2917 as members of a new genus did not provide enough detail. To make this clearer to the reader we have added the following statement to the results on line 182 (changes are underlined): “As noted by Adriaenssens and Brister (50), the Bacterial and Archaeal Viruses Subcommittee (BAVS) of the International Committee on the Taxonomy of Viruses (ICTV) considers phages sharing $\geq 50\%$ nucleotide sequence identity as members of the same genus. The closest V22 relative identified was *Vibrio* phage 1.063.O_10N.261.45.C7 (Genbank accession number: MG592441) sharing 32.9% average nucleotide identity (ANI); see Figure S2). Due to the lack of known myoviruses sharing an identity greater than 50% with V22, and the high ANI value (71.1%) found between V22 and GOV bin 2917, we have proposed both phages be considered members of a new genus called *Myoalterovirus* within the family *Myoviridae* (under ICTV review), featuring phage V22 as the type species.”

Furthermore, we also want to highlight the differences found between V22 and Enterobacteria phage T4. After comparison of both genomes at the NCBI using BLASTn, we verified that no significant similarity was found (overall identity values were below 1%). Also, when visual inspection of both genomes is performed, no synteny is detected throughout the whole genome. Then, we can conclude that phage T4 is distantly related to V22 and therefore they do not belong to the same genus.

Finally, we also would like to point out that we have been in contact with Dr. Evelien Adriaenssens (the Vice Chair of the BAVS of the ICTV), regarding the proposal of new genera. She informed us that, since 2016, they have been using whole genome nucleotide similarities, in combination with marker gene phylogenies to define genera. She also said that members of the same genus now typically share 70% nucleotide identity over the genome length with the type species genome and at a minimum 50% with all other members. We hope all this information helps to clarify this matter.

Major and minor concerns:

1. On lines 273, Results showed that GFP-gp26/gp27 can bind to a non-infected *A. macleodii* strain AD45 (Figure 6A) and the same for the phage absorption test (Figure 6B). The authors argued that since there is no CRISPR or prophage in the genome of AD45, thus it is not due to host immunity. This is not true since it is possible that AD45 has other types of anti-phage defense mechanisms such as RM systems or abortive infection systems.

We completely agree with the reviewer, who we thank for pointing out these other important phage defense systems. The reviewer refers to line 367 in the discussion, which we have now revised to read (changes are underlined): “Furthermore, no CRISPR-Cas system or prophages could be identified within the genome of AD45, suggesting the lack of infectivity is also not due to bacterial immunity or prophage-induced immunity. While other anti-phage defenses such as abortive infection or restriction modification systems could be responsible for terminating a potential V22 infection, it would be of great interest to find V22-like phages that do infect strain AD45 to then test if certain genetic elements, e.g., the AMGs carried by V22 and other marine phages (48), play critical roles in determining a successful phage infection.”

2. There are some concerns of the GFP contamination in the co-expression assay of Gp27 and

Gp26. In Figure 5, His-tagged GFP contaminants were both presented in GFP-gp26 and GFP-gp26/gp27, but the GFP protein varied quite a lot in two conditions.

Yes, we agree with the reviewer that there are apparent differences in the amount of free His-GFP purified between gp26 alone and gp26+gp27. Firstly, the gel in Fig. 5 shows proteins purified using Ni-NTA affinity chromatography, which rarely results in ultra-pure proteins. Furthermore, depending on the construct being expressed (e.g., gp23, gp24, gp26 and gp26/gp27) different ratios of these contaminants (such as free GFP) were observed, which can also be true for other proteins containing GFP tags. We also typically see this contaminant free His-GFP product when we have investigated other phage-encoded affinity proteins. For example, free His-GFP (~30 kDa) was also observed as a side product of GFP-tagged protein overexpression of a *Listeria* phage tailspike protein (gp15) (shown below) when using the same pQE30 vector system that was used in this study.

Due to the trimeric nature of phage tailspikes and tail fibers, we hypothesize that during folding there could be an issue with generating complete fibers due to the large size of GFP on the N-terminus as they fold. This could lead to physical cleavage to release His-GFP, or incomplete translation of the rest of the protein leading to the formation of free His-GFP. However, this hypothesis was not the aim of this paper, and we observe free His-GFP as a side product/contaminant that does not affect the results shown here. Importantly, as described and shown in Figure 7 and Supplementary Figure 4, we address the issue of this contaminant GFP product by using size exclusion chromatography to remove it completely from purified GFP-tagged gp26 proteins, which proves there is no difference in binding between Ni-NTA or SEC purified gp26 or gp26+gp27.

To better explain this to the readers, we have now revised line 267 (changes underlined): “As a side product, His-tagged GFP alone was present in all protein purifications, likely as a side product of over-expression as observed previously (Dunne et al. 2019); however, as shown below this did not affect our investigations.”

Figure 1D taken from Dunne et al. 2019. “Reprogramming Bacteriophage Host Range through Structure-Guided Design of Chimeric Receptor Binding Proteins.” *Cell Reports* 29 (5): 1336-1350.e4. <https://doi.org/10.1016/j.celrep.2019.09.062>. SDS-PAGE of purified full-length (FL) PSA GFP-gp15 (71.6 kDa) and of the GFP-gp15 C-terminal domain (CTD) (48.8 kDa). Notice the band at 30 kDa

On Line 538 "amounts indicated in figure legends". These amounts are not included in Figures and Figure legends.

Many thanks for observing this mistake. During initial microscopy experiments, we had used varying concentrations of proteins before deciding on only using 80 μg per assay. This statement slipped through during editing. In fact, we only show microscopy images for 80 μg mixed with cells in all figures. We have since corrected this in the results section (by removing mention of different protein amounts) and corrected the materials and methods as described below:

Line 270: "While none of the individual proteins (including GFP-gp26 alone) demonstrated cell binding (Fig. 5 A&C), to our surprise we observed complete cell wall decoration for GFP-gp26 co-expressed with gp27 (Fig. 5A)."

Line 538: "~~20 μg or 80 μg (amounts indicated in figure legends)~~ of GFP-tagged protein was added to the cells"

May 20, 2020

Prof. Francisco Rodriguez-Valera
Universidad Miguel Hernandez
Division de Microbiologia
Campus de San Juan
Apartado 18
San Juan de Alicante, Alicante 03550
Spain

Re: mSystems00217-20R1 (***Alteromonas myovirus V22* represents a new genus of marine bacteriophages requiring a tail fiber chaperone for host recognition**)

Dear Prof. Francisco Rodriguez-Valera:

Your manuscript has been accepted, and I am forwarding it to the ASM Journals Department for publication. For your reference, ASM Journals' address is given below. Before it can be scheduled for publication, your manuscript will be checked by the mSystems senior production editor, Ellie Ghatineh, to make sure that all elements meet the technical requirements for publication. She will contact you if anything needs to be revised before copyediting and production can begin. Otherwise, you will be notified when your proofs are ready to be viewed.

Sincerely,

David Knipe
Editor, mSystems

Journals Department
Table S5: Accept
Table S1: Accept
Fig. S2: Accept
Fig. S5: Accept
Fig. S1: Accept
Table S3: Accept
Fig. S3: Accept
Table S2: Accept
Fig. S4: Accept
Table S4: Accept